# Digital Health Interventions to Improve Adolescent HPV Vaccination: A Systematic Review

**DOI:** 10.3390/vaccines11020249

**Published:** 2023-01-22

**Authors:** Jihye Choi, Irene Tamí-Maury, Paula Cuccaro, Sooyoun Kim, Christine Markham

**Affiliations:** 1Department of Health Promotion and Behavioral Sciences, University of Texas Health Science Center at Houston, Houston, TX 77030, USA; 2Department of Epidemiology, Human Genetics and Environmental Sciences, University of Texas Health Science Center at Houston, Houston, TX 77030, USA; 3Institute of Health and Environment, Seoul National University, Seoul 08826, Republic of Korea

**Keywords:** adolescents, digital health, human papillomavirus, vaccination, interventions, parents, technologies

## Abstract

Digital technologies are being increasingly utilized in healthcare to provide pertinent and timely information for primary prevention, such as vaccination. This study aimed to conduct a systematic review to describe and assess current digital health interventions to promote HPV vaccination among adolescents and parents of adolescents, and to recommend directions for future interventions of this kind. Using appropriate medical subject headings and keywords, we searched multiple databases to identify relevant studies published in English between 1 January 2017 and 31 July 2022. We screened and selected eligible studies for inclusion in the final analysis. We reviewed a total of 24 studies, which included interventions using text messages (4), mobile apps (4), social media and websites (8), digital games (4), and videos (4). The interventions generally improved determinants of HPV vaccination, such as HPV-related knowledge, vaccine-related conversations, and vaccination intentions. In particular, text message and social media interventions targeted improved vaccine uptake behaviors, but little meaningful change was observed. In conclusion, digital health interventions can cost-effectively provide education about HPV vaccination, offer interactive environments to alleviate parental vaccine hesitancy, and ultimately help adolescents engage in HPV vaccine uptake.

## 1. Introduction

The burgeoning field of digital health, which includes categories such as mobile health (mHealth) and telemedicine, is at the forefront of healthcare transformations and public health efficacy. The ubiquitous and evolving nature of digital tools, including mobile and web-enabled devices, has led to a proliferation of technology-mediated interventions across the healthcare continuum, from prevention to post-treatment surveillance [1]. To improve health outcomes, these interventions facilitate access to real-time information, self-care management that allows patient autonomy, and enhanced remote communication with providers. Of particular significance is their impact on primary preventive services [2] such as vaccination. While child and adolescent vaccinations can reduce, and in some cases, eliminate many preventable diseases and provide additional health benefits (e.g., prevent costs associated with medical expenses), the success of any vaccination program ultimately depends on its ability to address parents’ vaccine-related attributes, such as forgetfulness and lack of knowledge, and parental decision-making about their children’s vaccine uptake [3,4]. In this regard, digital health interventions have demonstrated the potential to positively influence parental vaccination decisions and improve vaccine uptake among children [5]. They can track vaccination schedules, send reminders [6], counter misinformation to allay vaccine hesitancy [7], and provide dynamic online environments coupled with social media platforms that allow to share information with other parents and interact with vaccine experts [8].

Despite the growing number of digital health-based programs for child and adolescent vaccinations [9,10,11,12,13], there are comparatively fewer digital health interventions advocating for human papillomavirus (HPV) vaccination [14,15]. For example, among over 200 vaccination-related mobile smartphone applications (apps) currently available, only a small proportion pertains to HPV [16,17]. While HPV is responsible for approximately 4.5% (640,000 cases) of new cancer cases diagnosed worldwide [18], adolescent HPV vaccine uptake remains suboptimal compared to other vaccines [19]. This is primarily due to parents’ low awareness, vaccine safety apprehension, and unfunded concerns about HPV vaccination leading to an increase in sexual activity [20,21,22,23]. As eradication of vaccine-preventable diseases is a global public health priority [24], the promotion and uptake of the prophylactic HPV vaccine should be maximized by technology-mediated interventions. These interventions can efficiently educate parents with reliable information about HPV vaccination and offer interactive environments in which parental vaccine hesitancy can be alleviated [25]. Ilozumba et al. (2021) systematically reviewed digital interventions for HPV vaccination promotion for both adolescents and caregivers of adolescents, but the review was limited to only mHealth interventions delivered via mobile phones [16]. To broaden the scope, the present systematic review aimed to describe and assess the use of various digital tools for HPV vaccination promotion among adolescents and parents of adolescents and recommend directions for prospective digital health interventions to promote HPV vaccination.

## 2. Materials and Methods

A systematic review of the literature was conducted and reported using the guidelines of the Preferred Reporting Items for Systematic Reviews and Meta-Analyses (PRISMA) 2020 checklist [26]. The detailed protocol of this review was registered a priori with PROSPERO (CRD42022334244), an international database of prospectively registered systematic reviews [27].

### 2.1. Search Strategy and Information Sources

Eligible articles about digital health interventions for HPV vaccination promotion were reviewed extensively. The literature search was performed from the three major bibliographic databases of Medline Ovid, Embase, and Cochrane. With the conjunction “AND” and the disjunction “OR,” the following Medical Subject Heading (MeSH) terms and relevant keywords were used in the search process: child, adolescent, teenager, teen, youth, parent, guardian, Human Papillomavirus, HPV vaccination, HPV vaccine, intervention, digital, e-health, smartphone, mobile health, mHealth, mobile app, game, social media, text message, and web-based. Reference lists of selected articles from multiple journals were manually checked for other potentially relevant articles and all “related to” or similar articles of the identified articles were followed. In addition, a hand search of the grey literature (e.g., unindexed journals and relevant conference proceedings) was conducted. Titles, abstracts, and keywords were further screened for inclusion upon obtaining results that adhered to the search criteria.

### 2.2. Eligibility Criteria

Peer-reviewed articles written in English were included in this systematic review. Given the speed at which technology develops, the review aimed to explore the most recent research trends in digital health interventions for HPV vaccination promotion. Accordingly, the inclusion time period included articles that were published within the last 5 years, between 1 January 2017 and 31 July 2022. The review included articles specifically about the design, implementation, and evaluation of interventions that deliver information, increase awareness, and promote the uptake of HPV vaccination using digital technology that encompasses electronic tools, mobile devices, and web-enabled devices. Furthermore, the review included articles about digital health interventions that targeted exclusively adolescents, parents of adolescents, and parent-adolescent dyads. Articles were excluded from the review if they were observational studies, systematic reviews, meta-analyses, or content and sentiment analyses.

### 2.3. Study Selection Process

Study selection involved a four-step process. First, we systematically searched the databases using the MeSH terms and keywords. When selecting the studies, we independently reviewed the titles and abstracts of all records identified as relevant to this systematic review. Second, duplicate citations across the databases were identified using Endnote, manually verified, and subsequently removed. Third, the remaining abstracts were checked for eligibility. Fourth, the full-text articles of the included abstracts were screened according to the inclusion and exclusion criteria. This iterative selection and screening process was followed by a final synthesis of the eligible studies, which were qualitatively analyzed. This qualitative analysis entailed categorizing the interventions according to the technology used to deliver the interventions, the theoretical basis of the interventions, as well as the content, outcome measures, and results of the interventions. The same steps were taken with articles identified in the grey literature.

### 2.4. Quality Assessment for Risk of Bias

Quality assessments were performed to assess the methodological quality of included studies. We independently used the Cochrane Collaboration tool [28], which is used to make judgments about the extent of bias in randomized controlled trial (RCT) studies. RCT studies were evaluated for five potential biases: selection, performance, detection, attrition, and reporting. The component ratings were scored as low risk, high risk, or unclear. Furthermore, we followed the National Institutes of Health quality assessment tool for before-after (pre-post) studies with no control group (NIH) [29] to evaluate the quality of nonrandomized quasi-experimental studies. This tool comprised a total of 12 items and each item was rated as Yes, No or Not reported/Not able to determine/Not applicable. The report of the risk of bias assessment is discussed in the results section, and a full presentation is given in the Appendix A.

## 3. Results

### 3.1. Search Results

The literature search and review process are shown in Figure 1. We first searched Medline Ovid, PubMed, and Embase, retrieving a total of 1284 records. There were 692 records after duplicates were removed within and across databases. We further screened the title or abstract to identify whether the key terms were present, and whether the content fulfilled the inclusion criteria. We identified 65 records for full-text review and excluded 41 articles because they were not an intervention (*n* = 16), were not targeted to parents or adolescents (*n* = 11), were based on analyses of content or sentiment (*n* = 4), were not technology-based (*n* = 8), or had abstracts only (*n* = 2). We identified a total of 24 studies published between 1 January 2017 and 31 July 2022 as the final set of records for the review. No additional eligible articles were identified.

### 3.2. Study Characteristics

Among the 24 articles selected for analysis, four studies focused on text messaging [30,31,32,33], four studies pertained to mobile apps [34,35,36,37], eight studies were social media and website-based interventions [38,39,40,41,42,43,44,45], four studies were game-based interventions [46,47,48,49], and four studies were video-based interventions [50,51,52,53]. Thirteen studies (54%) targeted parents, seven studies (29%) targeted both parents and adolescents, and four studies (17%) targeted only adolescents. Fifteen of the studies [30,31,32,33,36,37,41,43,44,45,46,51,52,53] were RCTs and four studies [38,42,49,50] were non-randomized pre-post designs without a control group. Selected studies did not represent various geographical settings as most of the studies (18/24, 75%) were conducted in the United States. All but five studies were theory-informed; being the Health Belief Model and the Theory of Planned Behavior the most frequently employed theories. The main outcome measures examined in these studies were categorized as knowledge about HPV and HPV vaccination, vaccination intention, vaccine uptake, and qualitative feedback for those that tested a prototype of their interventions. Knowledge about HPV and HPV vaccination was measured in ten studies [31,36,37,38,39,40,46,47,49,51], vaccination intention in nine studies [36,37,38,42,44,50,51,53], vaccine uptake in 13 studies [30,31,32,33,39,40,41,42,43,44,45,46,52], and eight studies focused on qualitative user feedback of the interventions [34,35,39,42,46,47,48,49]. Common features across the interventions were dissemination of HPV and HPV vaccine information, tailored feedback or personal stories about vaccinating children against HPV, vaccine reminders, HPV discussion forums or Frequently Asked Questions (FAQ), scheduling assistance or clinic locator, and guidance on how to initiate conversations with children about HPV (Figure 2). Table 1 is an overview of the interventions organized by type of technology. A summary of the interventions is included in the Appendix A. We discuss the included interventions in detail below.

### 3.3. SMS Text Message Interventions

Four studies in this review were SMS text message interventions [30,31,32,33]. Text message reminders to parents of vaccine-eligible adolescents are generally reported as an effective means for their timely uptake of the HPV vaccine, especially regarding vaccine series completion [30,31,32,33]. In a study that assessed the effect of phone or text message reminders to parents, adolescents of parents who received text message reminders completed the series 71 days earlier compared to the control, with 18% more adolescents achieving full vaccination [30]. Although the time to receipt of subsequent doses for adolescents of participants in the phone reminder group was not different compared to the control, text message reminders were significantly effective for those who had already initiated the vaccine series. Another intervention found a contrasting result, where the benefits of text message reminders were restricted to individuals who had missed an earlier dose while those who were up to date with the vaccine almost always completed the three-dose schedule without the intervention [32]. A study conducted among an uninsured population also reported that adolescents of parents who received text message reminders returned for the second dose of the HPV vaccine at a higher rate than the control group. However, this finding was not statistically significant [31].

Text messages with tailored content in theory-informed interventions did not yield drastic success as anticipated. While timely text message reminders helped shift parents from the pre-contemplation and contemplation stages to preparation and fully vaccinated stages, educational information in the messages tailored to their stage of decision-making had minimal impact on their adolescents’ vaccine series completion [33]. In addition, no significant difference was observed in vaccination rates between motivational text messages emphasizing one’s susceptibility to HPV and self-regulatory text messages highlighting implementation intentions [32]. However, both types of messages elicited higher vaccination rates compared to the control condition. Three of the four studies reported that no gender differences were observed regarding the vaccine uptake of adolescents [30,31,32].

### 3.4. Mobile App Interventions

Four studies pertained to mobile apps developed to provide parents with information regarding adolescent HPV vaccination and promote uptake and completion of the HPV vaccine series [34,35,36,37]. Two of the studies focused on the formative assessment of a prototype HPV vaccine app, Vaccipack and Vax4HPV [34,35]. Distinct features of Vaccipack included a vaccine checklist, inspirational stories about parents drawing on common parental beliefs, and an active discussion forum about HPV vaccination. When the perceived usefulness of Vaccipack was measured, 82% of parents and 85% of adolescents found the app beneficial whereas a higher percentage of parents (88%) than adolescents (75%) showed intention to use the app. This finding was associated with higher levels of app acceptability among parents [34]. Users of Vax4HPV received personalized HPV recommendations tailored to their willingness to vaccinate their children, and assistance to search the nearest clinics. Users of this app were also provided an exemplar script to help them facilitate parent-child communication about the vaccine. In addition to acknowledging that the app closed the intention-behavior gap by offering scheduling appointments and locating clinics, parents suggested that the app provide gender-neutral information to eliminate vaccine hesitancy [35]. Education about adolescent vaccination and parent-adolescent communication about the HPV vaccine were the most cited anticipated benefits for both apps.

Two studies evaluated HPV vaccination intention and vaccine initiation behavior following the use of the apps [36,37]. HPV CancerFree (HPVCF) comprised various educational contents including debunking misinformation, functioned as a medium to facilitate parent-provider communication, and operated as an assistant to schedule HPV vaccination appointments. Although parents who used the app became significantly more knowledgeable about HPV and the HPV vaccine and perceived the app to be effective compared to those who did not use the app, HPV vaccination intention rates did not significantly differ between the two groups [36]. Consequently, vaccine initiation was not found to be significantly greater for parents who used the app compared to those who did not. Modules of Vacteens.org were similar to features in other apps, as they addressed misinformation about the HPV vaccine, provided a discussion forum between parents and providers, presented examples of parent-adolescent communication about the vaccine, provided clinic locators, and scheduling assistance for HPV vaccination. In this study, the rates of HPV vaccination initiation and series completion were 18.8% and 36.8% higher, respectively, among parents who used the app compared to those in the usual and customary information condition [37]. The app also bolstered parents’ positive HPV vaccine beliefs, which was a likely contributing factor for higher levels of vaccine initiation and series completion.

### 3.5. Social Media and Web-Based/Online Interventions

Eight interventions in this review were delivered via social media platforms and websites alike. In two interventions, which were a Facebook campaign to cultivate mother champions of HPV vaccination and a website providing mothers with tailored feedback from virtual assistants, both groups of mothers had significantly increased knowledge about HPV and HPV vaccine [38,39]. In the former study, mothers reported that the Facebook campaign addressed important gaps in knowledge about men’s susceptibility to HPV, as well as head and neck cancers and their link with HPV [38]. The majority of parents (86.4%) in this study noted that they gained confidence in starting conversations after their training as champions, which was consistent with the findings of another Facebook campaign, 3forME, that generated awareness and conversations among adolescents [39]. Likewise, adolescents who were notified every time a new message about HPV and vaccine-related facts was posted to a Facebook page, About Your Health, were more likely to have interpersonal discussions with others about what they learned and improved in knowledge at 3-months post-intervention [40]. However, the analyses did not reveal any significant differences by gender of adolescents for any of the outcomes.

Regarding vaccination behavior, a Facebook campaign, Health Chat, delivered to mothers was effective, as mothers reported significantly higher initiation of their adolescents’ HPV vaccination at 12- (71.3%) and 18-months (73.3%) posttest, compared to baseline (63.4%). Completion of HPV vaccination was conveyed by 62.5% and 65.9% of mothers at 12 and 18 months, respectively, both of which represented a significant increase compared to the baseline rate of 50.2% [41]. Another web-based intervention for parents accessed via a tablet resulted in a significant positive change as all participants intended to vaccinate their children and self-reported a high vaccination rate (95.8%) post-intervention [42]. Several other interventions exerted negligible effects on vaccination behavior. In the 3forME campaign, only two adolescents were vaccinated as a result of the intervention [39]. A different Facebook campaign had little or no impact among mothers in the upper socioeconomic status quartile and a negative effect among women in the lowest socioeconomic status quartile in terms of initiating vaccine uptake for their adolescents [43]. In the website-based intervention with the virtual assistants, no effects were found on HPV vaccination uptake although mothers experienced positive changes in the determinants of vaccination, such as knowledge, intention and perceived effectiveness of the vaccine [44]. Delivered on an iPad, the CHICOS website based on the concept of tailored messaging conveyed informational materials crafted to reflect parents’ HPV-related questions, experiences, and perceptions; there were no statistically significant differences in the improvement of vaccination intention or children’s vaccine completion between the intervention group and the control group [45].

### 3.6. Game-Based Interventions

Four studies illuminated the feasibility of digital games to increase knowledge about HPV and HPV vaccination and vaccine uptake among parents and adolescents [46,47,48,49]. For maximal engagement and motivation, these games had features to attract youth primarily by incorporating animated game characters with entertaining storylines. For example, the Land of Secret Gardens created a story about a secret garden as a metaphor for a preteen’s body and keeping it healthy [46]. FightHPV portrayed interactions between various characters such as epithelial cells, warts, precancerous cells, low-risk HPV, high-risk HPV, and HPV vaccine [47]. A common function of the included games was the communication of tailored feedback addressing knowledge about HPV, HPV-related diseases, and benefits of HPV vaccination via messages and episodes presented to players. A greater proportion of adolescents who played the Land of Secret Gardens game initiated the vaccine and had higher completion rates than their counterparts in the control group. However, these differences were not statistically significant [46]. Adolescents in the multimedia intervention noted higher intentions to discuss HPV vaccination with their parents after playing the game [49]. Adolescents who played the We Care-Teen game commented that they would like more interactions in the game to facilitate HPV-related discussions with parents and suggested to include HPV-related illness for both males and females [48].

### 3.7. Video-Based Interventions

Four video-based interventions for parents to improve HPV and HPV vaccine knowledge and intention to vaccinate their adolescents reported positive results [50,51,52,53]. In a digital story-telling intervention, participants watched two 3-min long personal digital stories about HPV vaccination from mothers of already vaccinated children. In this study, 74% of the participants intended to vaccinate their children, which significantly differed from 53% preintervention, and mothers expressed slightly more positive attitudes towards the need to vaccinate girls than boys after the intervention [50]. Is the HPV vaccine for me? Was a 6-min video mapping the adolescent HPV vaccine decision journey in Ireland with evidence-based information and reminders that most girls in the country have received the HPV vaccine [51]. Participants in the intervention group scored significantly higher on the knowledge assessment compared to the control group and 88% of participants indicated that the video increased the HPV vaccine acceptance for their adolescents, compared to 49% at the outset being undecided [51]. In another intervention, when parents were shown a video about the risks and benefits of the HPV vaccine or received reinforcement messages on a mobile tablet in a clinic examination room, their vaccine-eligible adolescents had up to 3-times greater odds of receiving a dose of the HPV vaccine [51,52]. Mothers in another video-based intervention received tailored messages addressing one versus five most common parental concerns about HPV vaccination. The adjusted mean scores for vaccination intent post-intervention were 3.5/10, 3.9/10, and 4.2/10 among the control, top-concern, and all-concerns groups, respectively. Compared to the control group, more mothers in the all-concerns group and the top-concern group reported “very high intent” to vaccinate (scores ≥ 8) by 7.9% and 1.9%, respectively [53].

RCT studies included in this review were at relatively low risk for selection and reporting biases, whereas detection bias was the most frequently occurring bias. Many of the studies did not provide clear and thorough information regarding their risk for performance and reporting biases. Regarding the single-group nonrandomized studies included in this review, they fulfilled most of the items listed on the assessment tool. However, none of the studies had a sufficiently large sample size to provide confidence in their findings. Moreover, none of the studies reported the blinding of outcome assessment or included outcome measures that were collected multiple times before and after the intervention. A full report of the quality assessment for each study is included in the Appendix A.

## 4. Discussion

### 4.1. Principal Findings

Over the last five years, HPV promotion interventions have been developed and delivered using various technologies: text messages, mobile apps, social media, digital games, and videos. Text messaging, as an early basic feature of mHealth [54], continues to be effective in reminding parents to initiate their adolescents’ HPV vaccination and to return for subsequent doses [55]. Mobile apps, which have advanced mHealth following the advent of tablets and smartphones, allow for the relay of educational components along with immediate feedback to increase parents’ HPV vaccination knowledge and intentions [56,57]. On social media platforms, important messages about HPV vaccination can be shared and parents are empowered to engage in vaccine-related public discourse [58]. Interactive social media interventions (e.g., online discussion forums, chat rooms, and “ask an expert” features [8]) can help build trust, combat vaccine bias, and ameliorate parents’ vaccine hesitancy [5,59]. However, a major caveat is that because the internet is a powerful conduit of vast amounts of vaccine arguments, anti-vaccination beliefs, and misinformation on social media may wield greater influence on vaccine-hesitant parents [60,61,62]. Recently, the use of digital games in a non-game context has gained popularity in public health to tackle vaccine acceptance [63,64]. Gamification features, such as audiovisual interfaces, interactive quizzes, reward systems, and unique characters, can be used to debunk myths about the vaccine in entertaining ways [65], and drive emotional engagement in players [66]. Game-based interventions may appeal to adolescents [67,68] and parents who are adept with technology and accept games to advocate controversial health issues such as HPV vaccination [69].

Our findings indicate that user perceptions and effects of the interventions were generally positive albeit with a few unexpected results. Contrary to the lack of theory-informed mHealth interventions reported by a previous review [16], almost all the interventions in this review were theory-based, which portrays an increasing trend of digital interventions using a theoretical framework. This finding is promising as researchers have emphasized applying theory to the design and evaluation of health promotion interventions to achieve desired health outcomes [70]. However, eight interventions in this review detected minimal merit of theory-based approaches [32,33,36,38,39,43,44,46]. For example, no differential effect was found between the messages incorporating a self-regulatory strategy and the messages using a motivational strategy. It was speculated that receiving a “plain” message without any theory basis may have been equally effective at increasing HPV vaccine uptake [32]. Furthermore, educational text messages to target the stage of parental vaccine decision-making referring to the Transtheoretical Model did not have added benefits [33]. In other interventions that employed theories, including the health belief model, social cognitive theory, theory of reasoned action, and inoculation theory [36,38,39,43,44,46], the interventions produced positive changes but not of statistical significance or they had only modest effects. Although not in line with earlier research that argued for theory-based interventions to increase effectiveness [71], our findings confirm the need to consistently re-evaluate theory-based digital health interventions to promote HPV vaccination.

While digital technologies are not supplanting traditional in-person medical consultations [72,73], several interventions in this review underscored their role as facilitators of effective provider-parent communication and education about HPV vaccination [35,37,46,48,52]. The interventions helped parents and adolescents become confident in discussing vaccination with their providers [37,40,48,52]. Moreover, parents could browse the apps to learn about HPV vaccination and make informed decisions, which lessened the reliance on providers. Education offered by the apps prior to clinic visits helped providers talk more comfortably with parents about HPV vaccination, knowing that many of their concerns were already covered [34,35,37]. For maximal effectiveness and safe use of digital interventions, providers’ insights about the credibility of these tools are crucial before dissemination to users. In fact, adolescents in Teitelman’s study were more willing to use an app and trust its content when it was advised by a healthcare provider [34], which elucidates that their acceptance of the app is conditional on the advice from their most trusted source of health information.

Another interesting point raised by this review is that although the interventions, particularly those delivered via apps, social media, and digital games, were well-received and strengthened key determinants of HPV vaccination, little meaningful change was observed in the actual vaccine uptake or vaccination rates [31,32,36,42,43,44,45,46]. The increase in HPV and vaccination knowledge, vaccine-related conversations, as well as positive vaccination beliefs and intentions are likely indications of the decision to vaccinate, but they may not be sufficient to cause behavior change, resulting in an intention-behavior gap [35,74]. Uncertainty may exist as to what psychosocial change is expected to turn parental decisional balance toward a decision to vaccinate [36]. Future digital interventions for HPV vaccination should reflect key differences between factors that influence intention versus actual vaccination behavior [75]. We anticipate that carefully crafted but succinct, resonating text messages can help parents and adolescents execute concrete action plans that can optimally meet the behavioral goal of HPV vaccination [76]. Novel approaches, such as digital games and social media platforms, have demonstrated great potential for promotion and educational purposes despite their nascent stages of development [77], but a more familiar and longstanding mHealth tool of text messaging is not to be underrated.

Except for five studies that targeted parents of daughters [37,41,43,44,51], the included studies targeted adolescents of both genders, given that vaccination rates for boys remain well below the Healthy People 2020 goals [52]. While no gender differences were observed regarding adolescents’ vaccine uptake and vaccine knowledge post-intervention [30,31,32,40], parents in one study were more likely to vaccinate girls than boys after the intervention [50]. Nonetheless, several studies in this review articulated the importance of gender-neutral HPV vaccination and providing information related to male susceptibility to HPV in the digital interventions to nudge boys to adopt the vaccination behavior [35,38,47,48]. Digital tools and social media platforms are increasingly exploited in the public health arena to increase awareness and acceptance of vaccination [78]. As part of this transformation, developers of digital interventions for HPV vaccination should consider tailoring design elements and content of the interventions to the gender of users to align with the recommended universal HPV vaccination [42,48].

Finally, this review affirmed the recurrently mentioned challenge of population appropriateness for digital health interventions. Despite their cost-effectiveness and convenience, digital interventions for vaccine adherence may not produce positive outcomes if they are implemented for populations of low socioeconomic status [79]. For example, we observed that an electronic reminder system with educational messaging about HPV and HPV vaccine did not significantly improve knowledge and vaccination rates among an underserved population [31], and a social media campaign was not effective among women of low socioeconomic status in initiating vaccine uptake for their adolescents [43]. These findings reveal a digital divide and the need to consider sub-population circumstances when implementing digital interventions for the controversial issue of HPV vaccination [16]. Trials of a prototype intervention and a subsequent evaluation are important for prospective parent users, given that users of technology tend to be younger [80]. For example, parent participants in five of the included interventions provided constructive feedback describing potential advantages and areas for improvement after their experience with the interventions [34,35,42,46,48]. As these parents had adolescents between the ages of 9 and 17 years, incorporating their opinions and proposed changes can gradually reduce the digital divide for older parents who may not readily accept such interventions.

### 4.2. Strengths and Limitations

There are several limitations in this systematic review. A limited number of databases were searched, which may have led to selection and reporting biases, and included studies were limited to distinct geographical settings, mainly in the US and European countries, although unintended. Social media interventions were limited to Facebook campaigns. The evidence in this review may also not be robust enough to represent all the current digital interventions for HPV vaccination because we focused only on parents and adolescents as the intervention recipients, without targeting providers. The strengths of this study are its focus on an overarching collection of digital tools and technology-mediated HPV promotion interventions, beyond mHealth. We were able to verify the advantages of using digital interventions but also their overlooked weaknesses. This review can also serve as a useful guide for countries with high digital technology penetration but low awareness of HPV vaccination to develop successful digital interventions for HPV vaccination, which can pave the path to adopt gender-neutral vaccination in the near future.

## 5. Conclusions

In this systematic review, we examined 24 digital health interventions incorporating text messages, mobile apps, social media and websites, games, and videos to promote adolescent HPV vaccination. Digital health interventions hold tremendous potential as a cost-effective and timely strategy to promote HPV vaccination; however, closing the intention-behavior gap remains a difficult task. While mobile apps and digital games facilitated education and active discussions around HPV thereby positively influencing knowledge and vaccination intentions, text message and social media interventions aimed to improve vaccine uptake behaviors, although the change was not always significant. Nonetheless, the interventions were found to support informed decision-making of adolescents and parents about HPV vaccination, expediting their communication with healthcare providers. Finally, continual revision of the interventions based on user feedback and considerations of parent characteristics is key to improving HPV vaccination outcomes among adolescents.

## Figures and Tables

**Figure 1 vaccines-11-00249-f001:**
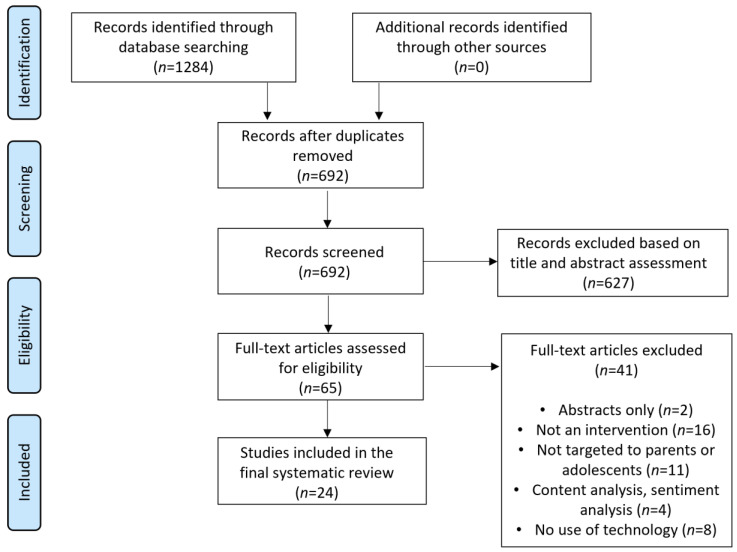
PRISMA flow diagram.

**Figure 2 vaccines-11-00249-f002:**
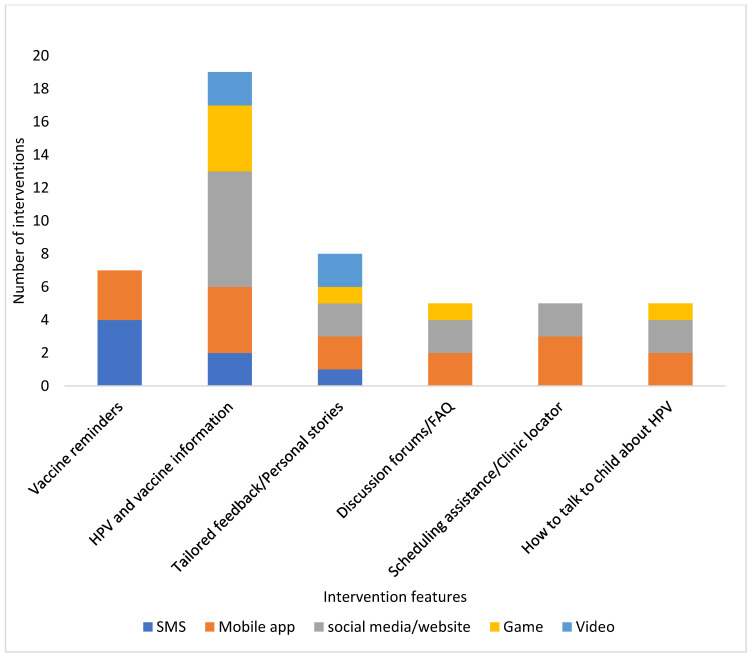
Features of the digital health interventions for adolescent HPV promotion by intervention type.

**Table 1 vaccines-11-00249-t001:** Organization of the studies by intervention type.

Intervention Type	Target	Theoretical Framework	Outcome Measures			
			Knowledge	Vaccination Intention	Vaccine Uptake	Qualitative Feedback
SMS Text message						
Rand et al., 2017 [30]	Parents	NA			X	
Richman et al., 2019 [31]	Parent-child dyads	NA	X		X	
Tull et al., 2019 [32]	Parents	HBM			X	
Wynn et al., 2021 [33]	Parents	TTM			X	
Mobile app						
Teitelman et al., 2020 [34]	Parent-adolescent dyads	IBM; TAM				X
Woodall et al., 2021 [37]	Parents and adolescents (daughters)	IDM, DIT	X	X		
Kim et al., 2022 [35]	Parents	Information systems research framework				X
Shegog et al., 2022 [36]	Parents	HBM; SCT; TRA	X	X		
Social media/websites						
Pot et al., 2017 [44]	Parents (of daughters)	IM		X	X	
Mohanty et al., 2018 [39]	Adolescents	HBM	X		X	X
Ortiz et al., 2018 [40]	Adolescents	HBM	X		X	
Dempsey et al., 2019 [45]	Parent	NA		X	X	
Chen et al., 2020 [42]	Parents and children	HBM; TPB		X	X	X
Buller et al., 2021 [41]	Parents (of daughters)	SCT; TT; DIT			X	
Chodick et al., 2021 [43]	Parents (of daughters)	Inoculation theory			X	
Sundstrom et al., 2021 [38]	Parents	HBM; TTM	X	X		
Game						
Amresh et al., 2019 [48]	Parent-adolescent dyads	SCT				X
Ruiz-Lopez et al., 2019 [47]	Adolescents	SCT	X			X
Cates et al., 2020 [46]	Parent-adolescent dyads	SDT	X		X	X
Occa et al., 2022 [49]	Adolescents	TPB; SCT	X			X
Video						
Dixon et al., 2019 [52]	Parents	TPB			X	
Chen et al., 2022 [50]	Parents	NA		X		
Marshall et al., 2022 [51]	Parent-adolescent (daughter) dyads	TDF	X	X		
Panozzo et al., 2020 [53]	Parents	NA		X		

HBM: Health Belief Model, TTM: Transtheoretical Model, IBM: Integrated Behavioral Model, TAM: Technology Acceptance Model, SCT: Social Cognitive Theory, TRA: Theory of Reasoned Action, IDM: Informed Decision Making, DIT: Diffusions of Innovation Theory, TT: Transportation Theory, TPB: Theory of Planned Behavior, IM: Intervention Mapping, SDT: Self-determination Theory, TDF: Theoretical Domains Framework.

## Data Availability

No new data were created or analyzed in this study. Data sharing is not applicable to this article.

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
