# Peer review of "Digital Health Interventions to Improve Adolescent HPV Vaccination: A Systematic Review"

_vaccines, 2023, doi:10.3390/vaccines11020249_

Round 1

Reviewer 1 Report

This paper describes a systematic review regarding digital interventions for the promotion of HPV vaccination for adolescents and parents. This is an interesting topic given the constantly evolving nature of digital health. However, there are some structural issues with the paper which need to be addressed. Please find my specific comments below.

Introduction:

‘largely depends on parents’ acceptance’- whilst acceptance is one factor which impacts the success of a vaccination program, the true measure is uptake and this is impacted by many things. This needs to be clearly explained as you include text message reminders in this review, and generally this intervention is not addressing acceptance.

“In this regard, digital health interventions have demonstrated the potential to positively influence parental acceptance of vaccination [5]. They can track vaccination schedules, send reminders- ref isn’t only about acceptance.” Again, I think you are confusing acceptance and uptake.

Methods:

Need to define what you mean by digital technologies for inclusion criteria.

Eligibility criteria: “not related to HPV vaccination”. Remove this from exclusion criteria, already covered in your inclusion criteria.

“Articles were also excluded if their interventions targeted providers or other intended users other than parents or adolescents.” This should be rephrased and added to the inclusion criteria.

Results:

“RCT studies included in this review were at relatively low risk for selection and reporting biases, whereas detection bias was the most frequently occurring bias. Many of the studies did not provide clear and thorough information regarding their risk for performance and reporting biases. Regarding the single-group nonrandomized studies included in this review, they fulfilled most of the items listed on the assessment tool. However, none of the studies had a sufficiently large sample size to provide confidence in their findings. Moreover, none of the studies reported the blinding of outcome assessment or included outcome measures that were collected multiple times before and after the intervention. A full report of the quality assessment for each study is included in Appendix A.” Move this section to the end of the results.

Figure 1- studies included in qualitative synthesis? It doesn’t look like you did a qualitative synthesis.

“Fifteen of the studies were RCTs and four studies were non-randomized pre-post designs without a control group”- this needs references.

“The main outcome measures examined in these studies were categorized as knowledge about HPV and HPV vaccination, vaccination intention, vaccine uptake, and qualitative feedback for those that tested a prototype of their interventions”- how many specifically of each?

“The most observed content across the interventions was HPV and HPV vaccine information followed by tailored feedback or personal stories, vaccine reminders, discussion forums or Frequently Asked Questions (FAQ), scheduling assistance or clinic locator, and guidance on how to initiate conversations with your child about HPV (Figure 1).” I’m not sure what this adds and this is also not related to figure 1.

Figure 2- need to label x and y axis

SMS interventions: is this based on four studies? Suggest mentioning how many studies at the start of this section before discussing results.

“Following the use of two of the interventions, a Facebook campaign to cultivate mother champions of HPV vaccination and”- I’m not sure what you mean by ‘following the use of the two interventions’

Discussion

Overall, the discussion needs to be better focused. What is the big ‘so what’ for this review? I.e what does it add to the current knowledge base. There is information in the discussion which is not mentioned in the results. This needs to be addressed.

Table 1 needs to be in the results not discussion

‘Digital health may fall short of completely supplanting in-person medical visits‘. I’m not sure how this is relevant.

“However, parents in these studies were more likely to vaccinate girls than boys after the intervention”- Not reported in the results.

“Despite their cost-effectiveness and convenience, digital interventions for vaccine adherence may exhibit different results if they are implemented for low-income and rural populations”- not mentioned in results

Conclusion:

The conclusion does not seem to link well with the results you have presented. This needs to be amended.

Author Response

Response to Reviewer 1 Comments

Point 1: ‘largely depends on parents’ acceptance’- whilst acceptance is one factor which impacts the success of a vaccination program, the true measure is uptake and this is impacted by many things. This needs to be clearly explained as you include text message reminders in this review, and generally this intervention is not addressing acceptance.

Response: Thank you for your comment. We have revised the sentence as the following:

While child and adolescent vaccinations can reduce, and in some cases, eliminate many preventable diseases and provide additional health benefits (e.g., prevent costs associated with medical expenses), the success of any vaccination program ultimately depends on its ability to address parents’ vaccine-related attributes, such as forgetfulness and lack of knowledge, and parental decision-making about their children’s vaccine uptake. (pg. 2)

Point 2: “In this regard, digital health interventions have demonstrated the potential to positively influence parental acceptance of vaccination [5]. They can track vaccination schedules, send reminders- ref isn’t only about acceptance.” Again, I think you are confusing acceptance and uptake.

Response: Thank you for your comment. We revised the sentence as the following:

In this regard, digital health interventions have demonstrated the potential to positively influence parental vaccination decisions and improve vaccine uptake among children. (pg. 2)

Point 3: Need to define what you mean by digital technologies for inclusion criteria.

Response: Thank you for your comment. We have revised the phrase in inclusion criteria as the following:

…using digital technology that encompasses electronic tools, mobile devices, and web-enabled devices. (pg. 3)

Point 4: Eligibility criteria: “not related to HPV vaccination”. Remove this from exclusion criteria, already covered in your inclusion criteria.

Response: Thank you for your comment. We have removed the phrase “and if the primary focus of the intervention was not related to HPV vaccination” from the exclusion criteria. (pg. 3)

Point 5: “Articles were also excluded if their interventions targeted providers or other intended users other than parents or adolescents.” This should be rephrased and added to the inclusion criteria.

Response: Thank you for your comment. We have rephrased the sentence and added to the inclusion criteria:

Furthermore, the review included articles about digital health interventions that targeted exclusively adolescents, parents of adolescents, and parent-adolescent dyads. (pg. 3)

Point 6: “RCT studies included in this review were at relatively low risk for selection and reporting biases, whereas detection bias was the most frequently occurring bias. Many of the studies did not provide clear and thorough information regarding their risk for performance and reporting biases. Regarding the single-group nonrandomized studies included in this review, they fulfilled most of the items listed on the assessment tool. However, none of the studies had a sufficiently large sample size to provide confidence in their findings. Moreover, none of the studies reported the blinding of outcome assessment or included outcome measures that were collected multiple times before and after the intervention. A full report of the quality assessment for each study is included in Appendix A.” Move this section to the end of the results.

Response: Thank you for your comment. We have moved the section above to the end of the results on pg. 10.

Point 7: Figure 1- studies included in qualitative synthesis? It doesn’t look like you did a qualitative synthesis.

Response: Thank you for your comment. We originally used the phrase adhering to the format of the PRISMA flow diagram, but we have rephrased it as “studies included in the final systematic review” in the diagram for clarity and to avoid confusion (pg. 4).

Point 8: “Fifteen of the studies were RCTs and four studies were non-randomized pre-post designs without a control group”- this needs references.

Response: Thank you for your comment. We have added references for the listed studies (pg. 4-5).

Point 9: “The main outcome measures examined in these studies were categorized as knowledge about HPV and HPV vaccination, vaccination intention, vaccine uptake, and qualitative feedback for those that tested a prototype of their interventions”- how many specifically of each?

Response: Thank you for your comment. Table 1 presents which studies focused on which outcome measures, as indicated by ‘X’ but we have added the number of studies that focused on each of the outcome measures along with the references for clarity. We have revised the sentence as the following:

Knowledge about HPV and HPV vaccination was measured in ten studies [31, 36-40, 46, 47, 49, 51], vaccination intention in nine studies [36-38, 42, 44, 50, 51, 53], vaccine uptake in thirteen studies [30-33, 39-46, 52], and eight studies focused on qualitative user feedback of the interventions [34, 35, 39, 42, 46-49]. (pg. 5)

Point 10: “The most observed content across the interventions was HPV and HPV vaccine information followed by tailored feedback or personal stories, vaccine reminders, discussion forums or Frequently Asked Questions (FAQ), scheduling assistance or clinic locator, and guidance on how to initiate conversations with your child about HPV (Figure 1).” I’m not sure what this adds and this is also not related to figure 1.

Response: Thank you for your comment. In addition to examining the characteristics of the interventions, such as their mode of delivery, target population, outcome measures and theoretical basis, we believe that it is important to also examine the common features of the interventions in the form of a brief content analysis. It can provide a snapshot of different features that have been incorporated in HPV vaccination promotion interventions to help parents’ decision-making about their children’s vaccine uptake. Figure 2 refers to this section describing the different features of the interventions. In addition, we have changed “contents” to “features.”

Common features across the interventions were dissemination of HPV and HPV vaccine information, tailored feedback or personal stories about vaccinating children against HPV, vaccine reminders, HPV discussion forums or Frequently Asked Questions (FAQ), scheduling assistance or clinic locator, and guidance on how to initiate conversations with children about HPV (Figure 2). (pg. 5)

Point 11: Figure 2- need to label x and y axis

Response: Thank you for your comment. We have labeled the axes, where X-axis presents the features of the digital health interventions for HPV promotion (“Intervention features”) and the Y-axis is the number of interventions (pg. 7).

Point 12: SMS interventions: is this based on four studies? Suggest mentioning how many studies at the start of this section before discussing results.

Response: Thank you for your comment. Similar to the rest of the sections, we have added the number of studies that focused on SMS interventions, which is as follows:

Four studies in this review were SMS text message interventions [30-33] (pg. 7).

Point 13: “Following the use of two of the interventions, a Facebook campaign to cultivate mother champions of HPV vaccination and”- I’m not sure what you mean by ‘following the use of the two interventions’

Response: Thank you for your comment. The phrase ‘following the use of the two interventions’ refers to the results of the preceding interventions (Facebook campaign and website). However, for clarity, we have revised the sentence as follows:

In two of the interventions, a Facebook campaign to cultivate mother champions of HPV vaccination and a website providing mothers with tailored feedback from virtual assistants, both groups of mothers had significantly increased knowledge about HPV and HPV vaccine [38, 39] (pg. 9).

Point 14: Overall, the discussion needs to be better focused. What is the big ‘so what’ for this review? I.e what does it add to the current knowledge base. There is information in the discussion which is not mentioned in the results. This needs to be addressed.

Response: Thank you for your comment. There are several key takeaways in this review. First, various digital technologies have been used to promote adolescent HPV vaccination in the last five years: text messages, mobile apps, social media and websites, digital games and videos. Second, most of the digital health interventions reviewed were theory-based, which supports the importance of applying theories to behavioral interventions to achieve desired health outcomes. Our findings revealed, however, minimal merit of employing a theoretical framework, suggesting that theory-based interventions need consistent re-evaluations. Third, while the interventions generally improved determinants of HPV vaccination, such as knowledge and vaccination intentions, the actual vaccine uptake behavior did not always significantly improve. This adds to the current knowledge base that determinants of a health behavior do not always result in the direct enactment of the behavior, resulting in an intention-behavior gap. Fourth, in this review, most studies on digital health interventions for HPV vaccination targeted adolescents of both genders and parents of both sons and daughters and reported user feedback that information related to male HPV vaccination should be provided, which aligns with universal HPV vaccination and combatting misconceptions about the HPV vaccine as a gender-specific vaccine. Lastly, this review highlights that digital interventions for vaccination promotion may not always result in positive outcomes and population appropriateness to accept technology-mediated interventions should be considered. This reflects issues such as digital divide and digital illiteracy that may be more prominent among underserved populations. Ultimately, this review shows that digital health interventions hold tremendous potential as opportunities to promote adolescent HPV vaccination in a cost-effective and timely manner, but there are some caveats to be cognizant of. We have revised and condensed the discussion section so that it is better focused, and is more clearly linked to the information presented in the results. We would appreciate your consideration.

Point 15: Table 1 needs to be in the results not discussion

Response: Thank you for your comment. We have moved Table 1 to the results section (pg.5-6).

Point 16: ‘Digital health may fall short of completely supplanting in-person medical visits‘. I’m not sure how this is relevant.

Response: Thank you for your comment. We wanted to make the point that, despite numerous benefits and positive impact of digital health interventions on healthcare, these technologies should not completely replace the need for patient contact as it is of paramount importance that providers communicate with parents and educate them about adolescent HPV vaccination via in-person medical visits. However, several studies in this review have demonstrated that digital health interventions can help mitigate some of the providers’ burden and responsibility and prepare parents and adolescents to a great extent before their encounters with providers. For clarity, we have amended the phrase as the following:  

While digital technologies are not supplanting traditional in-person medical consultations [72, 73], several interventions in this review underscored their role as facilitators of effective provider-parent communication and education about HPV vaccination [35, 37, 46, 48, 52]. (pg. 11).  

Point 17: “However, parents in these studies were more likely to vaccinate girls than boys after the intervention”- Not reported in the results.

Response: Thank you for your comment. We made the following revisions so that this section of the discussion addresses information that was mentioned in the results. First, we have indicated which studies targeted specifically female adolescents (daughters) and parents of daughters in Table 1 (pg. 5-6). Second, in the SMS text message section of the results, we have added that three of the four studies reported that no gender differences were observed regarding the vaccine uptake of adolescents, providing their corresponding references (pg. 8). Third, we have included in the results section (video-based interventions) the fact that mothers in one study expressed slightly more positive attitudes towards the need to vaccinate girls than boys after the intervention [50] (pg. 10). Lastly, the fifth paragraph of the discussion (pg. 12) describes how several studies expressed the need to include gender-neutral content in the digital health interventions. We already indicated in the results section that several studies [35, 38, 47, 48] emphasized gender-neutral HPV vaccination, and providing knowledge about male susceptibility to HPV and HPV-related illnesses for both genders. We have highlighted these points in red in the manuscript.

Point 18: “Despite their cost-effectiveness and convenience, digital interventions for vaccine adherence may exhibit different results if they are implemented for low-income and rural populations”- not mentioned in results

Response: Thank you for your comment. We reported in the results section that one Facebook campaign [43] had little or no impact among mothers in the upper socioeconomic status quartile and a negative effect among women in the lowest socioeconomic status quartile in terms of initiating vaccine uptake for their adolescents. We also reported in the results section that an electronic reminder system with educational messaging about HPV and HPV vaccine did not significantly improve knowledge and vaccination rates among an underserved population [31]. Based on these findings, we mentioned in the discussion that digital interventions for vaccine adherence may produce different outcomes if they are implemented for low-income and rural populations. For clarity, we have modified the phrase as the following:

Despite their cost-effectiveness and convenience, digital interventions for vaccine adherence may not produce positive outcomes if they are implemented for populations of low socioeconomic status (pg. 12).

Point 19: The conclusion does not seem to link well with the results you have presented. This needs to be amended.

Response: Thank you for your comment. We have revised the conclusion so that it better presents the results of this review. The revision is as follows:

In this systematic review, we examined 24 digital health interventions incorporating text messages, mobile apps, social media and websites, games and videos to promote adolescent HPV vaccination. Digital health interventions hold tremendous potential as a cost-effective and timely strategy to promote HPV vaccination, but closing the intention-behavior gap remains a difficult task. While mobile apps and digital games facilitated education and active discussions around HPV thereby positively influencing knowledge and vaccination intentions, text message and social media interventions aimed to improve vaccine uptake behaviors, although the change was not always significant. Nonetheless, the interventions were found to support informed decision-making of adolescents and parents about HPV vaccination, expediting their communication with healthcare providers. Finally, continual revision of the interventions based on user feedback and considerations of parent characteristics is key to improving HPV vaccination outcomes among adolescents (pg. 13).

Reviewer 2 Report

Title:  Digital health opportunities and interventions to improve adolescent HPV vaccine uptake: a systematic review

Author: Jihye Choi et al.

Recommendation:

In the submitted manuscript, the authors studied a Digital health opportunities and interventions to improve adolescent HPV vaccine uptake: a systematic review. In my opinion, the results presented in this paper are interesting and I suggest that the paper will be accepted for publication in the journal if the authors can make some revisions according to the following comments.

1-     The abstract should contain answers to the following questions: What problem was studied and why is it important? What methods were used? What are the important results? What conclusions can be drawn from the results? What is the novelty of the work and where does it go beyond previous efforts in the literature?

2-          They should double check the mathematical formulations, and also add appropriate references for governing equations.

3-    The English writing of the paper is required to be improved. Please check the manuscript carefully for typos and grammatical errors. Also, the English structure of the article, including punctuation, semicolon, and other structures, must be carefully reviewed.

---------------updated

Recommendation: In the submitted manuscript, the authors studied a Digital health opportunities and interventions to improve adolescent HPV vaccine uptake. In my opinion, the results presented in this paper are interesting and I suggest that the paper will be accepted for publication in the journal if the authors can make some revisions according to the following comments. 1. Change the title with meaningful and concise words. 2. The result section need mode modification. 3. Rearrange the abstract of the review with concise information. 4. Select a few keywords. 5. If possible include more literature articles from different sources for good compression and statistical analysis of the study. 6. The article's numbers (24) are not well enough to represent the importance of the study. 7. It would be better to supplement all sub-sections of the review with tables and graphs? 8. What does supplementary material mean? are there any supplementary materials available with this review? 9. The English writing of the paper is required to be improved. Please check the manuscript carefully for typos and grammatical errors. Also, the English structure of the article, including punctuation, semicolon, and other structures, must be carefully reviewed.

Author Response

Response to Reviewer 2 Comments

Point 1: Change the title with meaningful and concise words.

Response: Thank you for your comment. We believe that the title captures the content of this paper but for more conciseness, we can omit the words “opportunities” and “uptake” and change it to the following: Digital health interventions to improve adolescent HPV vaccination: a systematic review.

Point 2: The result section need mode modification.

Response: Thank you for your comment. We have revised the results section for a better flow and more clarity (pg. 4-10).

Point 3: Rearrange the abstract of the review with concise information.

Response: Thank you for your comment. Although the abstract is within the word limit according to the journal guidelines, we have revised the abstract for more conciseness and clarity. The revised abstract is as follows:

Abstract: Digital technologies are increasingly utilized in healthcare to provide pertinent and timely information for primary prevention, such as vaccination. This study aimed to conduct a systematic review to describe and assess current digital health interventions to promote HPV vaccination among adolescents and parents of adolescents, and recommend directions for future interventions of this kind. Using appropriate medical subject headings and keywords, we searched multiple databases to identify relevant studies published in English between 2017 and 2022. We screened and selected eligible studies for inclusion in the final analysis. We reviewed a total of 24 studies, which included interventions using text messages (4), mobile apps (4), social media and websites (8), digital games (4) and videos (4). The interventions generally improved determinants of HPV vaccination, such as HPV-related knowledge, vaccine-related conversations, and vaccination intentions. In particular, text message and social media interventions targeted improved vaccine uptake behaviors, but little meaningful change was observed. In conclusion, digital health interventions can cost-effectively provide education about HPV vaccination, offer interactive environments to alleviate parental vaccine hesitancy, and ultimately help adolescents engage in HPV vaccine uptake. (pg. 1).

Point 4: Select a few keywords.

Response: Thank you for your comment. We believe that all the keywords are relevant and appropriately represent the paper. The number of keywords is within what is suggested by the journal. However, we changed ‘HPV vaccination’ to ‘vaccination’ to avoid two-worded keywords.

Keywords: adolescents; digital health; Human papillomavirus; vaccination; interventions; parents; technologies

Point 5: If possible include more literature articles from different sources for good compression and statistical analysis of the study.

Response: Thank you for your comment. We conducted literature review and cited 25 articles to provide a succinct, yet informative background of this study. To conduct the systematic review, we searched three most comprehensive bibliographic databases for literature search in the health sciences. We also focused more on the qualitative synthesis and systematic analysis of the studies rather than statistical analyses, as this was not a meta-analysis reporting statistical results. We would appreciate your consideration.

Point 6: The article's numbers (24) are not well enough to represent the importance of the study.

Response: Thank you for your comment. We initially retrieved 1,284 records from the search of the databases and after meticulously screening the studies in several phases, the number of the studies included in the final review was 24. This is most likely due to having very tight, specific eligibility criteria, including studies published within the last five years and the fact that the use of digital technologies in the field of HPV vaccination may still be in nascent stages compared to other vaccines. We have also addressed this as a limitation of the study (pg. 13). We would appreciate your consideration.

Point 7: It would be better to supplement all sub-sections of the review with tables and graphs?

Response: Thank you for your comment. Because there were results unique to each intervention type, we provided a detailed overview of the findings of the reviewed studies. However, we have also attempted to illustrate key results of the review using tables, graphs and diagrams. We would appreciate your consideration.

Point 8: What does supplementary material mean? are there any supplementary materials available with this review?

Response: Thank you for your comment. The supplementary materials (Appendices A and B) are the risk of bias assessment of the studies and the summary of the reviewed interventions, which were submitted as separate documents at the time of manuscript submission to the journal.

Point 9: The English writing of the paper is required to be improved. Please check the manuscript carefully for typos and grammatical errors. Also, the English structure of the article, including punctuation, semicolon, and other structures, must be carefully reviewed.

Response: Thank you for your comment. We have carefully checked for any typos, grammatical errors and the structure in the revised manuscript.